

# Social Tripper

## Portal społecznościowy do organizowania i relacjonowania podróży

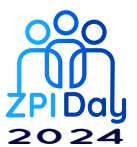

**Autorzy**: Leszek Kryzar · Wiktor Kubera · Szymon Szafraniec · Łukasz Wasilewski

**Opiekun:** Piotr Syga

### Streszczenie

Celem projektu inżynierskiego było opracowanie portalu społecznościowego, w którym możliwe jest organizowanie wspólnych podróży oraz dzielenie się relacjami za pomocą zdjęć i filmów. W ramach projektu stworzono dwie aplikacje: mobilną oraz webową, współpracujące z bazami danych, w których przechowywane są informacje o użytkownikach i wydarzeniach. Dodatkowo zaimplementowano system rekomendacji, w którym użytkownikom sugerowane są interesujące podróże oraz inni użytkownicy o podobnych zainteresowaniach. Do kluczowych funkcji aplikacji należy możliwość rejestracji użytkowników, tworzenia wydarzeń, uczestniczenia w nich oraz przeglądania relacji z wcześniejszych podróży. W aplikacji mobilnej możliwe jest śledzenie tras za pomocą GPS, rysowanie trasy na mapie oraz dodawanie zdjęć i filmów powiązanych z miejscami na trasie. Ostatecznym wynikiem projektu jest funkcjonalny portal społecznościowy, wspierający organizację podróży, umożliwiający dzielenie się doświadczeniami oraz tworzenie społeczności wokół wspólnych pasji. Dzięki innowacyjnemu podejściu do rekomendacji wydarzeń użytkownikom oferowane są bardziej interaktywne sposoby angażowania się w podróże.

## 1 WPROWADZENIE

### 1.1 Definicja problemu

Aplikacje typu social media od wielu lat są postrzegane jako czołowe narzędzia w branży rozrywkowej, a ich znaczący wpływ na kształtowanie trendów i opinii wśród użytkowników został wielokrotnie zauważony. Z jednej strony są uznawane za platformę umożliwiającą szybką i prostą wymianę wiedzy i informacji, z drugiej jednak wskazuje się na zagrożenia z nimi związane.

Coraz częściej są krytykowane za ich wpływ na zdrowie psychofizyczne, poczucie własnej wartości oraz sposób postrzegania rzeczywistości. Z biznesowego punktu widzenia rynek ten jest trudny do eksploracji, ponieważ jego uczestnicy charakteryzują się dużymi zasobami oraz mocno ugruntowaną pozycją.

Największe z aplikacji nie są specjalizowane w żadnym kierunku, co jest postrzegane zarówno jako ich wada, jak i zaleta. Uniwersalność i mnogość funkcjonalności pozwalają na łatwiejsze pozyskiwanie i utrzymywanie użytkowników na platformie. Jednak za pole do działania dla konkurencji uznawane są funkcjonalności specjalizowane, przystosowane do wykorzystania w bardziej profesjonalnym kontekście.

### 1.2 Cele projektu i oczekiwane korzyści

Za cel pracy przyjęto wytworzenie aplikacji (w wersji webowej oraz mobilnej), w której połączono cechy social media oraz narzędzi przeznaczonych do planowania podróży. Aplikacja umożliwia organizowanie podróży w nowoczesnym wydaniu, odpowiadającym wymaganiom ciągle rozwijającej się cyfrowej rzeczywistości.

Z założenia jest to platforma przyjazna użytkownikom niezależnie od ich wieku, języka komunikacji czy poziomu aktywności fizycznej. Użytkownicy niezaawansowani mogą dzięki niej czerpać inspirację i być aktywizowani w codziennym życiu. Dla weteranów sportu aplikacja stanowi niezawodne i intuicyjne narzędzie do organizowania wyjazdów oraz grup tematycznych.

Aplikacja umożliwia utrwalanie zarówno niesamowitych osiągnięć fizycznych, jak i codziennych aktywności, które wspierają budowanie kondycji oraz wdrażanie zdrowego trybu życia. System umożliwia poznanie osób o podobnych zainteresowaniach oraz wspiera rozwój lokalnej turystyki dzięki atrakcyjnym metodom promocji i relacjonowania wydarzeń.

## 2 PRZEGLĄD PRAC ZWIĄZANYCH Z TEMATEM I TECHNOLOGII

### 2.1 Przegląd istniejących rozwiązań i produktów konkurencyjnych

W dzisiejszym świecie aplikacji społecznościowych obserwuje się intensywną konkurencję. Wielkie platformy, takie jak Facebook, Instagram oraz Strava, charakteryzują się silnymi stronami, które postrzegane są zarówno jako zagrożenie, jak i szansa dla rozwijanej aplikacji.

**Facebook** - Jako jeden z najpotężniejszych portali społecznościowych charakteryzuje się szeroką gamą funkcjonalności, takich jak tworzenie grup tematycznych czy wydarzeń obejmujących praktycznie każdy temat. Niemniej jednak, grupy te są postrzegane jako mające ogólny charakter i nie zawierają specjalistycznych narzędzi dedykowanych określonym grupom użytkowników. Zaawansowane opcje organizowania wydarzeń w kontekście konkretnego rodzaju aktywności czy podróży nie są wspierane przez Facebook. W przeciwieństwie do wyspecjalizowanych platform, takich jak rozwijana aplikacja, zaawansowane narzędzia do planowania czy monitorowania wypraw nie są dostępne na Facebooku.

**Instagram** - Platforma ta jest głównie ukierunkowana na prezentowanie treści wizualnych, a dzielenie się momentami z podróży jest ułatwione, co przyciąga osoby zainteresowane estetyką oraz dokumentowaniem wypraw. Jednakże, w porównaniu z rozwijaną aplikacją, na Instagramie dostępnych jest o wiele mniej możliwości interakcji społecznych w kontekście organizowania wspólnych aktywności, co stwarza przestrzeń do wprowadzenia głębszych, bardziej zorganizowanych doświadczeń związanych z podróżami. Rozwijana aplikacja, skupiona na wspólnych wyprawach, towarzyskich doświadczeniach i integracji, jest postrzegana jako komplementarna alternatywa dla Instagramu, bez bezpośredniego konkurowania w zakresie wizualnej prezentacji treści.

**Strava** - Platforma, która została zdominowana przez użytkowników poszukujących narzędzi do monitorowania aktywności fizycznych, uznawana jest za silną konkurencję wśród aplikacji społecznościowych związanych z aktywnością na świeżym powietrzu, w tym podróżami. Działania Stravy są skoncentrowane na sportowcach i rywalizacji, oferując funkcje śledzenia wyników, tworzenia wydarzeń sportowych oraz klubów ukierunkowanych na osiąganie jak najlepszych rezultatów. W przeciwieństwie do Stravy, rozwijana aplikacja skierowana jest do podróżników poszukujących towarzystwa do wspólnych wypraw, w tym wielodniowych. Aplikacja nie koncentruje się na rywalizacji sportowej, lecz zapewnia integracyjne i społeczne doświadczenia, przyciągając osoby zainteresowane wspólnym podróżowaniem i poznawaniem nowych ludzi w ramach organizowanych wypraw.

### 2.2 Technologie i założenia projektowe

Najistotniejsze technologie, które użyliśmy w projekcie, to

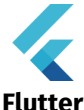 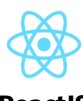 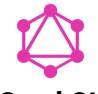 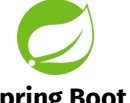 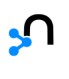 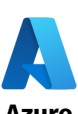

**Flutter**     **ReactJS**     **GraphQL**     **Spring Boot**     **Neo4j**     **Azure**

Aplikacja mobilna została opracowana w Flutter [8], który umożliwia tworzenie aplikacji na iOS i Android przy użyciu wspólnego kodu. Flutter wyróżnia się wydajnością i elastycznością, co czyni go lepszym wyborem w porównaniu do alternatywnych technologii, takich jak Xamarin czy React Native. [5]

Frontend został opracowany z wykorzystaniem ReactJS [9], popularnej biblioteki służącej do tworzenia dynamicznych aplikacji typu SPA [7]. ReactJS umożliwia szybki rozwój aplikacji oraz zapewnia rozbudowany ekosystem narzędzi.

Do komunikacji z backendem wybrano GraphQL [4], umożliwiający efektywne zapytania o konkretne dane, co poprawia wydajność aplikacji w porównaniu do tradycyjnego podejścia opartego na REST [1].

Backend oparty jest na Spring Boot [14], niezawodnej technologii Java, która ułatwia tworzenie REST API oraz integrację z innymi systemami.

Do przechowywania zależności między obiektami wykorzystano bazę Neo4j [12], grafową bazę danych, idealną do modelowania relacji w systemach społecznościowych [2].

Microsoft Azure [10] wybrano jako chmurową platformę wspierającą infrastrukturę, zapewniającą skalowalność, niezawodność oraz bezpieczeństwo.

W projekcie napotkano ograniczenia związane z czasem, zasobami i danymi. Krótkie terminy realizacji, szczególnie przy budowie platformy społecznościowej, ograniczyły czas na implementację, a część zasobów poświęcono na dokumentację. Trudności wystąpiły również przy zbieraniu danych do algorytmu rekomendacji, co wpłynęło na jakość wyników. Brak zasobów ludzkich, szczególnie w obszarze designu, oraz ograniczenia konta studenckiego na AWS wpłynęły na rozwój i wydajność systemu.

# 3    WYNIKI

## 3.1    Aplikacja Backendowa

Aplikacja backendowa została zrealizowana w postaci REST API zaimplementowanego w Spring Boot, w architekturze warstwowej. Dane aplikacji są przechowywane w dwóch bazach danych - relacyjnej (PostgreSQL) oraz grafowej (Neo4j). Takie podejście pozwala na wykorzystanie zalet obu technologii.

Baza relacyjna zapewnia integralność i bezpieczeństwo przechowywanych danych. Zapytania do bazy relacyjnej wykonywane są zarówno poprzez JPA, jak i funkcje oraz procedury składowane zdefiniowane w samej bazie, w przypadkach, gdzie narzut czasowy wynikający z wczytania danych z bazy i ich przetwarzania w JAVA byłby zbyt duży.

Grafowa baza danych umożliwia efektywne zamodelowanie sieci społecznościowej przy użyciu wyodrębnionych obiektów – węzłów w grafie oraz relacji między nimi – krawędzi w grafie. Dane z tej bazy pozyskiwane są przy użyciu zapytań w języku CYPHER.

Do zapisu multimediów generowanych przez użytkowników wykorzystano usługi chmurowe Azure Blob Storage. W bazie danych przechowywane są odnośniki do zasobów znajdujących się na zewnętrznym serwerze dostawcy chmurowego. Całość aplikacji została skonteneryzowana przy użyciu Dockera.

## 3.2    Aplikacja Webowa

Aplikacja webowa została zaprojektowana z wykorzystaniem nowoczesnych technologii, które zapewniają jej wydajność oraz płynność działania. Jednym z kluczowych rozwiązań zastosowanych w projekcie jest użycie GraphQL [4] do komunikacji z backendem. GraphQL jest językiem zapytań, który umożliwia pobieranie jedynie tych danych, które są rzeczywiście potrzebne w danym momencie. Dzięki temu uniknięto nadmiarowego przesyłania informacji, co przyspiesza czas odpowiedzi serwera oraz redukuje obciążenie sieci [1].

Kolejnym istotnym elementem wpływającym na wygodę użytkowania aplikacji jest zastosowanie React Router [16]. Dzięki tej technologii aplikacja działa w stylu Single Page Application (SPA), co eliminuje konieczność przeładowania całej strony przy każdej interakcji użytkownika. Zamiast tego, React Router umożliwia nawigację pomiędzy różnymi widokami w obrębie jednej strony, ładując jedynie niezbędne komponenty.

Takie podejście znacząco poprawia płynność i szybkość aplikacji, ponieważ użytkownik nie musi czekać na załadowanie całej strony po każdej akcji. Dzięki temu aplikacja stała się bardziej responsywna, a czas oczekiwania na interakcje użytkownika został zminimalizowany.

## 3.3    Aplikacja Mobilna

W aplikacji mobilnej stworzono interfejs umożliwiający zarządzanie wyprawami. Użytkownikom udostępniono funkcjonalność rozpoczynania wcześniej zaplanowanych wypraw z wykorzystaniem darmowego dostawcy usług mapowych, OpenStreetMap [3]. Aplikacja śledzi trasę użytkowników na mapie oraz umożliwia rejestrowanie zdjęć i filmów, które są automatycznie powiązane z określonymi punktami na trasie.

Tego typu funkcjonalność zwiększa interaktywność systemu, pozwalając na dzielenie się doświadczeniami w czasie rzeczywistym. Po zakończeniu wyprawy wszystkie pliki multimedialne oraz ścieżki są udostępniane w formie relacji, która stanowi jeden z kluczowych elementów aplikacji. Relacja zawiera suwak zdjęć i wideo, a obok wyświetlana jest mapa, przedstawiająca lokalizacje, w których zostały wykonane zdjęcia. Użytkownicy mogą przeglądać relację, korzystając z suwaka lub klikając pinezkę na mapie, aby obejrzeć udostępnione multimedia.

## 3.4    Algorytm Rekomendacji

W aplikacjach typu social media szczególny nacisk kładzie się na łączenie użytkowników poprzez sugerowanie interakcji oraz dostarczanie spersonalizowanych treści. Dzięki zastosowaniu algorytmu rekomendacji możliwe jest opracowanie systemu sugestii, który łączy osoby o wspólnych lub komplementarnych cechach. W efekcie użytkownikom umożliwia się łatwe poszerzanie sieci społecznościowej oraz korzystanie z treści dostosowanych do ich preferencji.

Prace nad stworzeniem algorytmu rekomendacji rozpoczęto od wyodrębnienia cech użytkowników uznanych za kluczowe dla wspólnego podróżowania i podejmowania aktywności fizycznej. Po przeanalizowaniu tej kwestii ustalono, że ludzie najczęściej wybierają do takich aktywności osoby tej samej narodowości lub posługujące się tym samym językiem, znajdujące się w podobnym wieku oraz charakteryzujące się zbliżoną kondycją fizyczną.

Kolejnym krokiem było pozyskanie danych użytkowników uwzględniających wcześniej wymienione cechy. Nie znaleziono jednak gotowych zbiorów danych spełniających oczekiwania, co doprowadziło do

podjęcia decyzji o wykorzystaniu danych ze Stravy [6] – serwisu będącego bezpośrednią konkurencją dla Social Tripper. Strava udostępnia system publicznych klubów, zawierających listy ostatnich aktywności członków. Dane pobierano z różnych klubów, najczęściej związanych z popularnymi markami, aby uzyskać jak największą różnorodność pod względem lokalizacji i rodzaju aktywności. Informacje dotyczące aktywności były ograniczone do nazwy użytkownika, jego lokalizacji oraz rodzaju wykonanej aktywności, jednak posłużyły jako baza do wygenerowania brakujących, kluczowych danych.

Pierwszym krokiem w generowaniu brakujących danych było określenie płci użytkowników na podstawie ich nazw. W tym celu wykorzystano API Namsor [11], które umożliwia wykonanie 5000 darmowych zapytań miesięcznie. Dzięki temu narzędziu, cechującemu się wysoką dokładnością, określono płeć użytkowników, nawet w przypadkach, gdy nazwa nie była typowym imieniem i nazwiskiem (często nazwy użytkowników Stravy zawierały pseudonimy, emotikony lub nazwy własne). Uzyskana informacja o płci została wykorzystana nie tylko jako cecha wzbogacająca model, ale także jako podstawa do generowania kolejnych parametrów.

Kolejnym krokiem było wygenerowanie wieku oraz wskaźnika BMI [18] dla użytkowników. BMI, w uproszczony sposób, pozwala na oszacowanie kondycji fizycznej człowieka. Do tego celu wykorzystano publiczne dane WHO [17] z 2016 roku, zawierające średnie wartości wskaźnika BMI w zależności od kraju i płci, oraz dane World Population Review [15] z 2023 roku, przedstawiające medianę wieku mieszkańców poszczególnych krajów w podziale na płeć. Na podstawie tych danych oraz przy użyciu rozkładu normalnego, wiek oraz wskaźnik BMI zostały losowo przypisane użytkownikom.

Ostatecznie w algorytmie rekomendacji wykorzystane zostały kluczowe parametry użytkowników, obejmujące wiek, wskaźnik BMI, kraj, płeć oraz listę rodzajów aktywności.

Po zebraniu wszystkich danych stworzono przypadki do trenowania i testowania modeli. Wytworzono 250 przypadków zawierających losowe pary użytkowników oraz ocenę w skali od 0 do 1, określającą stopień dopasowania i zasadność wzajemnej rekomendacji użytkowników. Przy ocenie par uwzględniono różne czynniki.

Największy wpływ na ocenę przypisano lokalizacji. Użytkownicy z tego samego kraju otrzymywali automatycznie co najmniej połowę punktów. W przypadku użytkowników z krajów posługujących się tym samym językiem, sąsiadujących ze sobą lub podobnych kulturowo, przyznawano mniej punktów niż parom z tego samego kraju, lecz nadal znaczną ilość. Założono, że osoby z różnych krajów rzadziej wyruszą razem na wycieczkę, ale mogą być zainteresowane wzajemną obserwacją, zwłaszcza jeśli są w podobnym wieku i preferują zbliżone aktywności.

Wpływ wieku na ocenę wynosił około 20% – im mniejsza różnica wieku, tym więcej punktów przyznawano. Podobny wpływ miały rodzaje aktywności. Maksymalna liczba punktów przypadała na pary z identycznymi aktywnościami, średnia liczba punktów – na aktywności częściowo pokrywające się lub pokrewne, a minimalna – na całkowicie różne rodzaje aktywności. Wskaźnik BMI miał minimalne znaczenie w przypadku osób z różnych krajów, lecz znacznie większe w przypadku par z tego samego kraju, ponieważ różna kondycja fizyczna mogła stanowić problem przy bardziej wymagających wyprawach. Płeć oceniano marginalnie, bez istotnego wpływu na wynik.

Warto zaznaczyć, że wszystkie przypadki oceniono subiektywnie. Generowanie ocen przy pomocy uprzednio stworzonego wzoru uznano za niewskazane, ponieważ eliminowałoby potrzebę stosowania modelu uczenia maszynowego.

Kolejnym etapem było przekształcenie przypadków treningowych i testowych na macierze cech liczbowych oraz wybór odpowiednich modeli. Ze względu na charakter problemu, którym była regresja, zdecydowano się na zastosowanie lasu losowego z biblioteki Scikit Learn oraz na stworzenie sieci neuronowej z wykorzystaniem biblioteki PyTorch.

Przekształcenie danych dotyczących wieku i aktywności przebiegło stosunkowo prosto. Aktywności zostały pogrupowane w siedem ogólnych kategorii, a następnie rozdzielone na siedem kolumn wskazujących, czy dany użytkownik wykonywał określony rodzaj aktywności. Większym wyzwaniem okazało się przetworzenie lokalizacji. Rozbicie jej na 195 kolumn (gdzie w każdym przypadku tylko jedna zawierałaby prawdę) uznano za nieefektywne. W tym celu zastosowano algorytm GloVe [13], umożliwiający uzyskanie reprezentacji wektorowych dla słów.

Algorytm GloVe działa na podstawie zagregowanych globalnych statystyk współwystępowania słów w korpusie tekstowym, a uzyskane wektory odzwierciedlają liniowe podstruktury w przestrzeni wektorowej słów. Do reprezentacji lokalizacji wykorzystano wektory o wymiarze 50, ze względu na ich szybkość generowania, ograniczony wpływ na zwiększenie rozmiaru modelu oraz najlepsze wyniki w późniejszych testach regresji według wybranych metryk.

Przypadki podzielono w stosunku 80:20, co odpowiada 200 przypadkom treningowym oraz 50 testowym. Na początkowym etapie testy przeprowadzono na macierzy cech liczbowych, w której naprzemiennie umieszczono kolumny odpowiadające użytkownikowi 1 i użytkownikowi 2. Wyniki uzyskane po dostosowaniu optymalnych hiperparametrów przedstawiono poniżej.

| Model | max_depth | n_estimators | min_samples_split | Kolumny połączone |
|---|---|---|---|---|
| Las losowy | 12 | 70 | 2 | Nie |

Tabela 1: Parametry pierwszego testu lasu losowego

Metryki: Średni błąd kwadratowy - 0.075, Współczynnik determinacji - 0.307

| name_1 | age_1 | bmi_1 | location_1 | gender_1 | activities_1 | score |
|---|---|---|---|---|---|---|
| Catherine Brampton | 37 | 22.4 | United Kingdom | female | ['Walk'] | 0.77 |
| Stephanie Leonard | 39 | 28.4 | USA | female | ['Walk'] | 0.55 |
| Martin Arnholz | 42 | 28.1 | France | male | ['Walk'] | 0.04 |
| **name_2** | **age_2** | **bmi_2** | **location_2** | **gender_2** | **activities_2** | **pred_score** |
| Fitrial Kamal | 38 | 29.3 | Indonesia | male | ['Run'] | 0.492568 |
| Miguel A O Z. | 35 | 24.2 | Mexico | male | ['Ride'] | 0.542396 |
| Adrian Borma | 46 | 19.3 | Romania | male | ['Ride'] | 0.209179 |

Tabela 2: Fragment wyników pierwszego testu lasu losowego

| Model | Wielkość warstw ukrytych | Współczynnik uczenia | Optimizer |
|---|---|---|---|
| Sieć MLP | [64, 32] | 0.000004 | RMSprop |
| **Liczba epok** | **Skalowanie cech** | **Kolumny połączone** | |
| 17000 | Nie | Nie | |

Tabela 3: Parametry pierwszego testu sieci neuronowej

Metryki: Średni błąd kwadratowy - 0.076, Współczynnik determinacji - 0.303

| Model | Wielkość warstw ukrytych | Współczynnik uczenia | Optimizer |
|---|---|---|---|
| Sieć MLP | [64, 32] | 0.000004 | RMSprop |
| **Liczba epok** | **Skalowanie cech** | **Kolumny połączone** | |
| 17000 | Tak | Nie | |

Tabela 4: Parametry drugiego testu sieci neuronowej

Metryki: Średni błąd kwadratowy - 0.048, Współczynnik determinacji - 0.557

| name_1 | age_1 | bmi_1 | location_1 | gender_1 | activities_1 | score |
|---|---|---|---|---|---|---|
| Catherine Brampton | 37 | 22.4 | United Kingdom | female | ['Walk'] | 0.77 |
| Stephanie Leonard | 39 | 28.4 | USA | female | ['Walk'] | 0.55 |
| Martin Arnholz | 42 | 28.1 | France | male | ['Walk'] | 0.04 |
| **name_2** | **age_2** | **bmi_2** | **location_2** | **gender_2** | **activities_2** | **pred_score** |
| Fitrial Kamal | 38 | 29.3 | Indonesia | male | ['Run'] | 0.697849 |
| Miguel A O Z. | 35 | 24.2 | Mexico | male | ['Ride'] | 0.602572 |
| Adrian Borma | 46 | 19.3 | Romania | male | ['Ride'] | 0.036639 |

Tabela 5: Fragment wyników drugiego testu sieci neuronowej

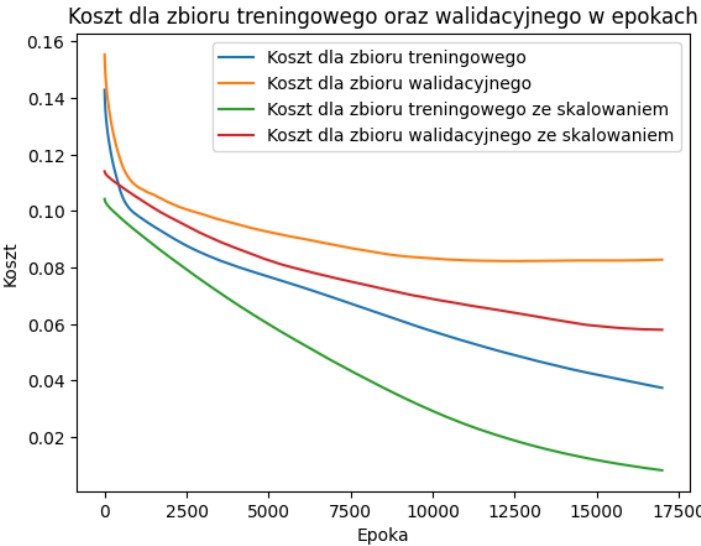

Rysunek 1: Krzywa uczenia modelu sieci neuronowej

Model sieci neuronowej, uczony na skalowanych cechach, uzyskał zdecydowanie najlepsze wyniki na przypadkach testowych. W tym momencie pojawiło się pytanie, jak sieć zachowa się przy predykcji dla rzeczywistej osoby. Przeprowadzono test, w którym jako pierwszego użytkownika wykorzystano dane jednego z członków zespołu, a następnie dokonano predykcji oceny rekomendacji dla każdego użytkownika pobranego ze Stravy (około 1000 przypadków). Zmieniono jedynie lokalizację pierwszego użytkownika, czyli kluczową cechę, w celu obserwacji zmian wyników.

| name_1 | age_1 | bmi_1 | location_1 | gender_1 | activities_1 |
|---|---|---|---|---|---|
| Łukasz Wasilewski | 22 | 20.0 | Poland | male | ['Walk', 'Hike'] |

Tabela 6: Parametry użytkownika 1 do predykcji sieci neuronowej

| name_2 | age_2 | bmi_2 | location_2 | gender_2 | activities_2 | score |
|---|---|---|---|---|---|---|
| Tapio Katava | 40 | 29.2 | Finland | male | ['Walk'] | 0.9864378 |
| Gi Chan Kim 1963 | 24 | 25.2 | South Korea | male | ['Walk'] | 0.93857765 |
| Charalampos Giavris | 39 | 33.4 | Greece | male | ['Walk'] | 0.9310318 |
| Murakami Taishi | 46 | 25.3 | Japan | female | ['Walk'] | 0.9258576 |
| George Hall | 25 | 33.9 | Australia | male | ['Run'] | 0.9134602 |

Tabela 7: 5 najwyżej rekomendowanych użytkowników

| name_2 | age_2 | bmi_2 | location_2 | gender_2 | activities_2 | score |
|---|---|---|---|---|---|---|
| Steve Ferg | 22 | 18.0 | USA | male | ['Run'] | 0.7909001 |
| Clayton Eck | 28 | 19.5 | USA | male | ['Run'] | 0.78721935 |
| Nate Apathy | 28 | 22.1 | USA | male | ['Run'] | 0.7825834 |
| Luis Santamassino | 33 | 20.8 | USA | male | ['Run'] | 0.781996 |
| Adrian Gonzalez | 36 | 19.4 | USA | male | ['Run'] | 0.78196883 |

Tabela 8: 5 najwyższych rekomendacji ze zmienioną lokalizacją na 'USA'

Model wykazał dobre wyniki dla użytkownika z USA, jednak rekomendacje dla użytkownika z Polski wydają się być dość losowe. Szczególnie niezrozumiała jest prawie maksymalna ocena dla użytkownika z Finlandii, z którym jedyną wspólną cechą jest aktywność "Walk"oraz płeć "male". Najprawdopodobniej wynika to z dużej liczby użytkowników z USA (około 300) w danych Stravy, w porównaniu do stosunkowo niewielkiej liczby użytkowników z Polski (15), co sprawia, że model generuje lepsze predykcje dla osób z USA.

Aby poprawić działanie algorytmu, podjęto próbę połączenia kolumn obu użytkowników, co miało umożliwić modelom lepsze wykrywanie zależności między cechami. W tym teście, zamiast podawać wiek użytkownika 1 i 2 osobno, model otrzymywał różnicę wieku, różnicę BMI i inne cechy. Oto wyniki testów:

| Model | max_depth | n_estimators | min_samples_split | Kolumny połączone |
|---|---|---|---|---|
| Las losowy | 20 | 70 | 2 | Tak |

Tabela 9: Parametry drugiego testu lasu losowego

Metryki: Średni błąd kwadratowy - 0.050, Współczynnik determinacji - 0.539

| name_2 | age_2 | bmi_2 | location_2 | gender_2 | activities_2 | score |
|---|---|---|---|---|---|---|
| Kamil Romanowski | 24 | 21.2 | Poland | male | ['Walk'] | 0.960428 |
| Anna Fedyszyn | 35 | 22.7 | Poland | female | ['Walk'] | 0.770000 |
| Bernadeta Stecyk | 28 | 30.0 | Poland | female | ['Walk'] | 0.769000 |
| JARRYD DANIEL ZOGHBY | 23 | 28.4 | RSA | male | ['Walk'] | 0.647571 |
| Hans-peter Stockmann | 25 | 33.0 | Germany | male | ['Hike'] | 0.634142 |

Tabela 10: 5 najwyższych rekomendacji lasu losowego z danymi użytkownika z Polski 6

| Model | Wielkość warstw ukrytych | Współczynnik uczenia | Optimizer |
|---|---|---|---|
| Sieć MLP | [64, 32] | 0.000004 | RMSprop |
| **Liczba epok** | **Skalowanie cech** | **Kolumny połączone** | |
| 37500 | Nie | Tak | |

Tabela 11: Parametry trzeciego testu sieci neuronowej

Metryki: Średni błąd kwadratowy - 0.064, Współczynnik determinacji - 0.411

Model wykazał dobre wyniki dla użytkownika z USA, jednak rekomendacje dla użytkownika z Polski wydają się być dość losowe. Szczególnie niezrozumiała jest prawie maksymalna ocena dla użytkownika z Finlandii, z którym jedyną wspólną cechą jest aktywność "Walk"oraz płeć "male". Najprawdopodobniej wynika to z dużej liczby użytkowników z USA (około 300) w danych Stravy, w porównaniu do stosunkowo niewielkiej liczby użytkowników z Polski (15), co sprawia, że model generuje lepsze predykcje dla osób z USA.

Aby poprawić działanie algorytmu, podjęto próbę połączenia kolumn obu użytkowników, co miało umożliwić modelom lepsze wykrywanie zależności między cechami. W tym teście, zamiast podawać wiek użytkownika 1 i 2 osobno, model otrzymywał różnicę wieku, różnicę BMI i inne cechy. Oto wyniki testów:

# 4 WNIOSKI

## 4.1 Podsumowanie wyników

W ramach projektu opracowano jednolitą platformę, umożliwiającą zrzeszanie miłośników podróży oraz budowanie społeczności wokół pasji do odkrywania nowych miejsc. Jednym z kluczowych osiągnięć była implementacja funkcjonalności tworzenia relacji z wypraw, która zapewnia użytkownikom łatwy dostęp do wspomnień poprzez interaktywne połączenie zdjęć, wideo i tras na mapie. Zastosowany algorytm rekomendacji umożliwia dostarczanie spersonalizowanych treści, co zwiększa zaangażowanie użytkowników oraz umożliwia odkrywanie wydarzeń dopasowanych do ich preferencji i zainteresowań.

W ramach platformy stworzono także interaktywną mapę, która w czasie rzeczywistym obrazuje przebieg trwającej podróży, dostarczając użytkownikom wizualnie atrakcyjnych treści. Funkcjonalność publikowania postów umożliwia łatwe dzielenie się materiałami z wybraną grupą odbiorców, co wspiera interakcje między użytkownikami oraz rozwój społeczności. Wszystkie te elementy stanowią kompleksowe rozwiązanie wspierające planowanie, dokumentowanie i dzielenie się doświadczeniami z podróży w sposób nowoczesny i angażujący.

## 4.2 Najważniejsze osiągnięcia projektu

Najważniejszym sukcesem projektu było stworzenie innowacyjnej platformy społecznościowej, która integruje funkcje planowania, rejestrowania i dzielenia się doświadczeniami z podróży. Wyróżniającą cechą jest interaktywny interfejs mapowy, umożliwiający użytkownikom śledzenie przebytej trasy w czasie rzeczywistym oraz przypisywanie multimediów do konkretnych lokalizacji. Rozwiązanie to znacząco wzbogaca sposób dokumentowania wypraw, czyniąc go bardziej wizualnym i angażującym.

Kolejnym kluczowym osiągnięciem było opracowanie algorytmu rekomendacji, który dostarcza spersonalizowane treści, zwiększając wartość użytkową systemu. Funkcjonalność tworzenia relacji z podróży pozwala użytkownikom na łatwe archiwizowanie wspomnień i dzielenie się nimi w sposób interaktywny, stanowiąc unikalny element platformy.

Projekt połączył zaawansowaną technologię z intuicyjnym designem, tworząc przestrzeń dla miłośników podróży do wymiany doświadczeń i budowania społeczności. Taki kompleksowy system stanowi znaczący krok w kierunku nowoczesnych rozwiązań w obszarze mediów społecznościowych dedykowanych podróżnikom.

## 5 KIERUNKI ROZWOJU

Jednym z głównych kierunków rozwoju aplikacji będzie dalsze udoskonalanie algorytmu rekomendacji, który ma oferować użytkownikom bardziej spersonalizowane sugestie. Planuje się poszerzenie zakresu rekomendacji, aby obejmowały również interesujące miejsca do odwiedzenia oraz grupy. Taki rozwój umożliwi użytkownikom odkrywanie nowych celów podróży, aktywności oraz społeczności podróżniczych, a także pozwoli na bardziej świadome planowanie wypraw.

W przyszłości aplikacja mogłaby zostać wzbogacona o integrację z technologią streamingu, co umożliwiłoby użytkownikom transmitowanie na żywo swoich wypraw, wydarzeń czy aktywności na świeżym powietrzu. Taka funkcjonalność pozwoliłaby podróżnikom i uczestnikom wydarzeń na dzielenie się doświadczeniami w czasie rzeczywistym z innymi członkami społeczności, wzbogacając interaktywność platformy.

Kolejnym pomysłem jest integracja z urządzeniami wearables, takimi jak smartwatche czy opaski fitness, w celu monitorowania aktywności użytkowników w czasie rzeczywistym i dostarczania spersonalizowanych informacji na temat zdrowia i kondycji podczas podróży. Funkcje te obejmowałyby analizę wyników fizycznych, zarządzanie trasami, a także udostępnianie osiągnięć i wyników w postaci wyzwań, które mogłyby motywować innych do aktywności.

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
