# OpenReview forum: "SocialTripper"
_pwr.edu.pl/Wrocław_University_of_Science_and_Technology/2024/ZPI_Day — Wrocław University of Science and Technology 2024 ZPI Day Submission_

### Official Review · Reviewer_kRjX · 2024-12-03
**Too much focus on recommendation**

**Confidence:** 5
**Significance Of Results:** 2
**Overall Quality:** 3

**Compliance With Template:**

3: Average Quality – The article includes most of the required sections, but some may be incomplete, written in a general or unclear manner. The content is correct but requires further refinement.

**Description Of Results:**

3: Average Quality – The results are described with moderate detail. Some examples or evaluation elements are present but insufficiently developed or incomplete.

**Feedback On Consistency:**

The general description is consistent, but the text focuses too much on the recommendation module and analysis of its work, istead of the base application. It is unknown if the application is functional in terms of real users, or even functionalities. Instead we are given explanation how to mix random data in a random form to get results, which are nothing more than random - from this we see that the recommendation module works for random data. We don't know if its good for users, and we don't know anything about the rest of the application (web or mobile).

**Potential For Development:**

Hard to say if the application is even finished, but authors point out to some possible future steps.

**Project Nature Evaluation:**

Technical description is started correctly, but not finished. Instead the focus is on semi-analysis of random data. Which was not the point of the project. As such we do not know if the main mobile and web application were finished, or if it only recommendation module.

**Technical Language Precision:**

3: Average Quality – The language is mostly appropriate but may contain minor terminological or stylistic errors. Some statements might lack precision or require improvement for better readability.

---

### Official Review · Reviewer_g1gz · 2024-12-05
**Social Tripper - Portal społecznościowy do organizowania i relacjonowania podróży**

**Confidence:** 4
**Significance Of Results:** 5
**Overall Quality:** 5

**Compliance With Template:**

4: High Quality – The article contains all the required sections, which are well-written and substantively correct, although minor errors or shortcomings may be present. The overall structure is clear and coherent.

**Description Of Results:**

4: High Quality – The results are described in detail and supported by usage examples or evaluations. The description is reliable but may lack full depth of analysis.

**Feedback On Consistency:**

The size of the paper exceeds the one assumed in the paper preparation rules twice. Nevertheless, it the paper is definitely worth reading its extended contents. The paper contains not only rich technological characteristics but also information about the recommendation algorithm implemented by the team, which is based on newral network. The project is very ambicious, so it was almost impossible to describe it in shorter form. However, the quality verification of the product is limited to the recommendation algorithm – demanding one with the high risks of errors, but not the only such area of the system.

**Potential For Development:**

A big advantage of the paper is careful and relatively detailed (relevant to the required size of the paper) description of motivation for the technology stack choice. It is woth noticing that the technologies are not the ones which are very popular among students. The choice of the technologies is not only motivated very well but it is also very good – appropriate to the character of social media applications with stress to severa important aspects of the product.

**Project Nature Evaluation:**

Evaluate the nature of the project. Does the project exhibit characteristics of an engineering work, such as the level of utility, application of technical methods, and technological solutions?

**Technical Language Precision:**

5: Very High Quality – The language is entirely appropriate for a technical report. All terms are used correctly and precisely, and the style is professional, clear, and coherent, without any errors or ambiguities.

---

### Official Review · Reviewer_bZSG · 2024-12-05
**SocialTripper, ZPI2024 review**

**Confidence:** 5
**Significance Of Results:** 5
**Overall Quality:** 4

**Compliance With Template:**

4: High Quality – The article contains all the required sections, which are well-written and substantively correct, although minor errors or shortcomings may be present. The overall structure is clear and coherent.

**Description Of Results:**

4: High Quality – The results are described in detail and supported by usage examples or evaluations. The description is reliable but may lack full depth of analysis.

**Feedback On Consistency:**

Artykuł jest spójny, autorzy uwzględniają wszystkie wymagane sekcje a ich kolejność jest logiczna. Rozdziały i sekcje wzajemnie się uzupełniają pozwalając śledzić tok prac autorów.
Pewnym problemem może być, nietypowe dla języka technicznego, stosowanie długich zdań wielokrotnie złożonych. Może powodować to utrudnienia w zrozumieniu, natomiast nie wpływa na ogólną jakość artykułu. Wartym uzupełnienia w sekcji 1.1 jest podanie cytowań do wskazanych twierdzeń (obwinianie social mediów) oraz doprecyzowanie i uzupełnienie źródła twierdzeń odnośnie trudności eksploracji i ugruntowanej pozycji istniejących graczy. Dodałoby to artykułowi cechę weryfikowalności.

W kwestii spójności artykułu, warto rozważyć przearanżowanie rozdziału 3.4 być może wydzielenie sekcji Testy, natomiast z pewnością reorganizacja umiejscowienia tabel oraz komentarza (analizy) tych tabel. Mniej znaczącą uwagą jest tu fakt zwyczajowego opisywania (caption) tabel nad nimi. Sam opis powinien też wyjaśniać jednostki użyte w tabelach. Nie jest wskazane mieszanie języków w artykule i warto byłoby zamiast zmiennych użyć tłumaczenia na język polski (lub zapewnić takie w podpisie). W ramach testów, warto uwzględnić też ewaluację aplikacji webowej oraz mobilnej pod względem zgodności z WCAG czy wrażeniami użytkowników. Wobec istotności algorytmu rekomendacyjnego oraz jego obszernych testach, takie testy zgodności mogłyby być opisane krótko, jedynie potwierdzając ich wykonanie i wynik.

**Potential For Development:**

Autorzy wskazują 3 możliwe kierunki rozwoju, krótko opisując swoje plany w każdym z punktów. Szczególnie interesującym i wskazującym na innowacyjne podejście jest ostatni z wskazanych pomysłów - wykorzystanie przenośnych czujników. Integracja zarówno aplikacji mobilnej, jak i internetowej ze smartbandami, smartwatchami i innymi lekkimi urządzeniami może pozwolić na wykorzystanie luki w istniejących mediach społecznościowych.

**Project Nature Evaluation:**

Praca wykazuje się typowymi cechami inżynierskimi. Autorzy wskazują jasne zastosowanie oraz wskazują konkurencję. Wobec stwierdzeń autorów o zdominowaniu rynku oraz trudnościach wdrożenia nowych projektów, aplikacja przed wdrożeniem może wymagać znacznych środków przeznaczonych na promocję oraz podkreśleniem zalet (i ciągłym rozwojem, co autorzy również zauważają w rozdziale 5) algorytmu rekomendacji, który wydaje się być istotnym wyróżnikiem. Sam artykuł również ma charakter techniczny, inżynierski. W zdecydowanej większości artykuł jest precyzyjny, przedstawia mierzalną weryfikację osiągnięć oraz opisuje wykorzystane narzędzia.
Dobrym uzupełnieniem (kosztem pustych lub prawie pustych linii) mogłoby być wzbogacenie opisu sekcji 3.4 o odpowiednie wzory i schematy).
W artykule warto też podać link do wersji internetowej aplikacji, a najlepiej repozytorium z kodem źródłowym.

**Technical Language Precision:**

4: High Quality – The language is appropriate for a technical report. Terminology is used correctly, and statements are precise, with only minor shortcomings that do not affect the overall clarity.

---

### Decision · Program_Chairs · 2024-12-10

Accept (Poster)